# From Ancient Patterns of Hand-to-Hand Combat to a Unique Therapy of the Future

**DOI:** 10.3390/ijerph20043553

**Published:** 2023-02-17

**Authors:** Artur Kruszewski

**Affiliations:** Department of Individual Sports, Jozef Pilsudski University of Physical Education in Warsaw, 01-813 Warsaw, Poland; artur.kruszewski@awf.edu.pl

**Keywords:** agon, innovative agonology, kalokagathia, martial arts, Olympic movement, safe fall, self-defence art

## Abstract

The purpose of this publication is to provide generalized knowledge of the area of changes that took place over past centuries in relation to health, social and cultural conditions. In Greek mythology, it was necessary to nurture both body and spirit to be a perfect human being. This link between the concepts of physical beauty and goodness can be also found in later works dedicated to ancient Greek history. Particularly in Greek myths, and in Greek education in general, it was believed that both physical and spiritual excellence were necessary to raise men to achieve their true form. Some of the main forms of implementing this idea were hand-to-hand combat exercises (wrestling, boxing and pankration). Ideas characteristic of the world of ancient Greece, in a general sense, can be observed in the culture of the Far East. The main difference is the fact that these principles did not survive in Western culture as a result of transformation into a consumer society focused on the rejection of moral principles. The brutalisation of the forms of the Roman Games meant that the ideals of the ancient world were forgotten for more than 1500 years. The modern Olympic Games were resurrected in the 19th century. Inspired by the ancient Greek cult of health of body and spirit, they gave rise to a movement known as Olympism. In the Olympic Charter written by Coubertin, Olympism was called “a philosophy of life exalting and combining in a balanced whole the qualities of body, will and mind”. The combat sports disciplines have had their place there since the beginning of the modern Olympic Games. The evolution of hand-to-hand combat disciplines, including numerous scientific studies indicating a very broad impact in the area of health, led to this type of physical activity being reached for as an essential element in supporting the health-promoting behaviour of society. Nowadays, physical activity in the area of exercise with elements of hand-to-hand combat, combat sports, or martial arts is an indissoluble link in the prevention and treatment of 21st century diseases. For Parkinson’s disease patients, drug treatment is an essential resource for continuing to function in society, but it will not be completely effective without supporting the treatment with appropriate and attractive physical activity (e.g., “Rock Steady Boxing”). Of similar importance is the prevention of dangerous falls, which are common in this population as well as among the elderly or those affected by other diseases of civilisation. Implementing the principles and techniques of safe falling in the teaching of the young population significantly increases the likelihood of applying appropriate responses to these individuals in adulthood and old age. Actions that should be taken now for prevention can be implemented through social programmes, such as “Active today for a healthy future”.

## 1. Ancient Greek Games

The term “martial arts” is basically used to describe the totality of hand-to-hand-fighting systems that originated in Eastern cultures. The origins of martial arts can be legitimately linked to three phenomena: the healing arts (e.g., acupuncture), the art of self-discovery (e.g., yoga) and life skills (e.g., meditation). However, martial arts originated in all parts of the world. Some of the oldest accounts are available in writings documenting the culture and daily life of the ancient Greeks [1].

There is a noticeable close connection between knowledge passed down through generations and the search for new solutions through the use of simple stories that create the principles of social cooperation. In general terms, they are commonly known as myths. These stories had a big influence on how Greek thought developed and how it gave rise to science. The principles of general conduct were defined by referring to mythical figures, which are unsurpassed models of conduct. Therefore, the ancient Greeks created rules of social conduct based on the works of Homer and Hesiod, and the pattern of their conduct was the Iliad. According to Greek mythology, maintaining one’s physical and spiritual health is essential for becoming a perfect human. This essential reference related to the culture of the body and mind was permanently enshrined in the later writings of ancient Greece.

In addition, with regard to the ancient Olympic Games, the values and principles underlying them are rooted in myth. Heracles, who instituted the Olympic Games, introduced the olive crown as a prize in honour of Zeus to thank him for helping him complete his fifth labour. In its essence, Olympic myths at the time also had four basic functions: cosmological, historical, sociological and psychological [2]

In ancient Greece, a theory of agonism was created that emphasized the positive aspects of certain forms of struggle. This theory was justified by science, philosophy, religious norms, and ethical principles. All of these forms of collective consciousness perceived physical culture in different aspects, and their perceptions of physical fitness were different. Widespread intellectual and emotional acceptance made sport an attribute of the Hellenistic way of life: a naked, oil-gleaming, well-built body distinguished a Greek from a barbarian, as did language and religion.

“Hellenism” can be understood as a term for the modern cultural construction and philosophical perspective that dominated this part of the world for nearly three centuries. Therefore, it is believed that the ancient Greeks created a civilisation that valued artistic practice and whose core were the ideals of democracy [3].

In Ancient Greek culture, the approach to care of the elderly attracted great attention of the state, philosophers, artists, writers and physicians. Ancient Greek medicine is also notable, including elder care, as an independent field of medicine. Hippocrates described the course and treatment of typical geriatric diseases and methods of treatment. Approaches to the state’s public pension system and to caring for veterans of legionnaires led to the creation of “homes for the elderly” through land grants. Attention should be paid to the valuation of old age that is passed on by tradition in myths. In the Greek tradition, old age and the elderly are presented and evaluated in a very positive way, and age is associated with wisdom. Respect and support for parents and grandparents was deeply rooted in morality and even regarded as a sacred duty [4].

The love of the great thinkers of that period for the art of sports, which was remembered in the image of a naked athlete, is widely known. Plato, Aristotle, Xenophon and others discussed the beauty of athletes, which they noticed not only in its external form, but also associated it with goodness [5,6].

Greek mythology is full of heroic figures distinguished by their wrestling skills—Zeus, Theseus and Odysseus. Heracles became a mythical personification of the sport of wrestling, and wrestling and pankration competitors were called “Heracles”. It was assumed that the creators of ancient sports training were mythical heroes: Peleus, Theseus and Heracles [7].

The presentation of beauty in the naked body of an athlete is a unique reference of Greek culture, indicating the perception of beauty not only in its external form, but also—and perhaps above all—as an ideal expression of a certain kind of soul. Concepts of beauty have evolved throughout history and are culture-specific. In the Western world, a concept that has been associated with beauty is kalokagathia, an ideal that combines the physical beauty and moral worth of human beings. The ideal of a harmonious combination of bodily beauty and moral goodness was thereby one of the main ideals. In particular, people from the upper classes were attributed with the combination of beauty and goodness.

Heracles as an outstanding personality of the ancient world, gaining fame thanks to numerous victories in sport and war, became for his citizens the image of the will to immortality. The life of Heracles inspired the ancient Olympic Games, and the symbol of the sacrificial pyre is one of the clearest examples of this. Probably the first ancient Greek sports were organized as sacrificial rituals [8]. Over time, along with the development of sports disciplines, lighting the Olympic flame was a reference to the flame of the “sacrifice pyre” as a sign of sacrifice. In performing this ritual, the greatest person, the victor, was sacrificed to the gods in a symbolic sacrifice. The greatest honour of the victor was, therefore, the possibility of making a symbolic sacrifice to the gods, not something of material value [9].

The winners of the competition at the Olympic Games, like their mythological idols, set the best example for the younger generation to shape a beautiful body and do good. The competition was so concentrated that winning once was very difficult, and winning repeatedly was remarkable.

Despite the widespread acceptance of male-only participation in the Games, we also find the most unexpected victors there. The Spartan woman Cynisca twice won the chariot competition, her trophy being a horse trainer. Among the great winners, there is also Leonides of Rhodes, who won 12 individual victories. Herodoros of Megara won the trumpet competition nine times in a row. Two wrestlers won the boys’ competition followed by five more wins in the open competition [10,11].

In 708 BC, wrestling was the most important element of the newly introduced pentathlon discipline. It was the most popular sport at that time. The winner of the pentathlon was probably determined through multiple eliminations; to win, one participant had to win at least three out of five events, including wrestling.

Fist fighting became an Olympic sport in 688 BC. These fights were very fierce, and the competitors used soft leather protectors that they wrapped around their hands before the start of each duel. The famous Melancomas of Caria (1st century CE) was able to win fights without striking, exhausting his opponents instead. In the final stage of the ancient Olympic Games, the most brutal competition was introduced. Pankration was a discipline that combined boxing and wrestling. Pankration fights, as they allowed the possibility of injuring the opponent, led to an increase in the brutality of the fights, even causing the death of one of the participants. Despite this, or maybe because of this, the Greeks loved them [7,8,9].

## 2. Roman Games—Gladiatorial Combat and the Ancient Olympic Games

The birth of the Roman Games is linked to the Etruscan custom of honouring the dead through combat rather than human sacrifice. Gladiatorial combat in Rome probably first appeared in 264 BC at the funeral of the senator Junius Brutus Pera. Six warriors (among the gladiators were also two sons of an aristocrat) fought a battle to honour the late. This event caused a very rapid rise in popularity and became one of the main Roman pastimes. The systematic increase in the wealth of Roman citizens and the conquest of new territories meant that numerous slaves were brought to Italy. By the end of the third century BC, slaves were used at the funerals of aristocrats, although still honouring the memory of the deceased, usually a few dozen captives each.

Over time, gladiatorial combat (munera) became more of an “advertisement” for its organisers than an actual tribute to the deceased. Whereas the fight was once watched in silence only by the deceased’s family, it was now mostly a crowd of random people, curious about the bloody spectacle and loudly “cheering” on the gladiators. Gradually, the religious ritual turned into entertainment for the people.

In 183 BC, the family of the late Publius Licinius Crassus organised battles involving as many as 200 captives. The fights stretched over several days and gathered crowds of spectators. From then on, gladiatorial combat became the Games.

The Games, established in the Roman Empire, were already, in essence, focused only on satisfying the shallow expectations of the citizens, and the very procedure of maintaining and training gladiators became a mere business from which one could make a fortune. Spectators valued the gladiators’ physical abilities, their fighting skills or tactical prowess, overlooking the values of goodness and moral attitudes. It was then that the famous slogan “bread and circuses” was coined.

The Games were organised by the emperor, and the number of festivals during which they were held increased. During the reign of Octavian Augustus, the total number of days for the Games was 66 days a year; later, it was more than a hundred, and in the fourth century AD, it was 175, if one adds the fights organised by the emperor and various rich men. It was usually the weaker gladiators who first stood up to fight. They were staged in groups in pairs, a few from each school. Then, exotic animals were used, and they fought among themselves or with the gladiators. Particularly in demand were huge Germanic aurochs, bears, African tigers, lions and panthers, which were imported from the remotest parts of the Empire. At the end, there was the main fight of the day, in which the two best gladiators from two different schools faced off. Some gladiators were promoted by arranging for them to fight easily defeated opponents.

The gradual demoralisation of the rules for running the Games resulted in increasingly brutalised fights. Organisers outdid themselves with ideas to attract more interest from the crowds. The Games became increasingly macabre, with a culmination in the middle of the first century AD. In addition to the usual fights between armed gladiators on equal footing, monstrous executions were carried out in front of the audience; the condemned were set on fire alive, crucified or thrown to the lions to eat.

The limits of cruelty were increasingly crossed; the emperor Domitian, known for his cruelty and black humour, had a taste for fights between women and dwarfs. All boundaries were crossed by the demented Commodus, who personally duelled in the arena.

In 107 AD, Emperor Trajan organised the largest Games in history: they lasted 123 days, and 10,000 gladiators fought in them [12].

The popularity of the Games was threatened in the 4th century AD when a large proportion of Romans professed Christianity. The idea of this religion could not be reconciled with the bloody Games, which were still quite popular. Gladiatorial combat was first banned by Emperor Constantine the Great in 326 AD at the insistence of the Church. Thus began the decline of the Roman games, which, through successive edicts, banned soldiers and Roman officials from participating in fights, recruited gladiators as bodyguards and led to the abolition of most gladiatorial schools in 399 AD. The last official fights in Rome took place in 404 AD [13].

Rome’s new position as a world power and their general relative social wealth resulted in the emergence of new diseases appearing as mass phenomena. The old virtues of moderation and frugality turned into greed and addiction to pleasure. Thus, the Roman people, under the banner of prosperity, degenerated into a society of leisure, consumption, fun and a throwaway mentality [14].

## 3. Modern Olympic Games

After 393, no Games were held for more than 1500 years. The modern Olympic Games were resurrected in the 19th century by an idealistic French man, Pierre de Coubertin. He had read about the ancient Greek Olympic Games and wanted to revive them. The ancient ideas impressed him so much that he believed the Olympic Games could contribute to world peace. Coubertin, in introducing the modern Olympic Games, believed that morality, purity, honesty and sporting selflessness were as important and as highly valued as muscular strength. According to Coubertin, Olympism, as a broader idea and through sports competition, was to combine the even development of moral qualities with the development of physical fitness [15].

The Games, inspired by the ancient Greek cult of health of body and spirit, gave rise to a movement called Olympism, the meaning of which was given in 1914 by the first Olympic Charter, which included the basic principles and rules of the Olympic movement. In this historic document written by Coubertin, Olympism became “a philosophy of life exalting and combining in a balanced whole the qualities of body, will and mind” [16].

Olympic philosophy is largely based on humanist anthropology, such as that developed by the ancient Greek philosophers. The development of the Olympic movement built around Olympic values, such as equal opportunities or the principles of “fair play”, gave rise to the creation of the Olympic philosophy [17,18].

Among the great ideas of the contemporary, modern world, the Olympic ideal, through its multidimensionality, is one of the best expressions of our identity. In its reference, it allows one to identify with this idea not only as a European, but also global, uniting the people and citizens of the world. The uniqueness of this ideal, supported by a spectacle enjoying worldwide interest, lies in its worldwide character.

However, the ideology of Olympism did not escape criticism. It was criticized for the elitism and racism of its first period, which inherently contradicted the mission of universal ideals and equal opportunities for all people and all nations. The discourse conducted by the main proponents of the idea of Olympism had to lead to changes in the commonly known ideology. These changes mainly satisfied various interests of social groups and movements. A clear example of such changes is the gradually increasing participation of women in Olympic sport and the expansion of the Olympic program to disciplines previously associated only with men’s competition. The second example would be the transformation of the initial Olympic movement, in which only aristocrats were allowed to participate, into a movement that took into account the needs of the working class. Such decisions were probably caused by the social and political events taking place in the interwar period and ensured the possibility of extending the influence of the movement to wider social groups [19].

Aiming at the goal of popularizing Olympic competition, which was proposed by Pierre de Coubertin, he sought to transform Olympism into a global idea. The basic reference was constant accessibility, opening the possibility to compete to anyone who meets the requirements of Olympism. Referring to the ancient patterns, he wanted to create a movement that would unify entire societies in its nature, and by directing interests to sports rivalry, it would increase the chances of maintaining peace. The Olympic Movement, evaluating it in retrospect, showed significant resourcefulness in promoting the assumptions that were the basis for its creation [20,21].

### Hand-to-Hand Combat in the Modern Olympic Games

Hand-to-hand combat (boxing and wrestling) was on the programme of the Games proposed by Coubertin. Wrestling and boxing events were held at all Summer Olympics since their introduction to the programme at the 1904 Summer Olympics, with the exception of the 1912 Summer Olympics in Stockholm, where no boxing tournament was held because Swedish law banned the sport at the time. The popularity of combat sports around the world developed in tandem with changes resulting from reforms within the IOC. New combat sports of Asian origin joined the family of Olympic disciplines. The first was judo (as a demonstration discipline on the programme of the 1964 Tokyo Olympics, and officially from the 1972 Munich Games onwards). Taekwondo was featured since the 2000 Sydney Olympic Games. During the Tokyo 2020 (held in 2021) Olympic Games, karate (including the Paralympic Games programme) was included as a demonstration discipline.

In a sense, an extension of the mission of ancient wrestling was the emergence of judo on the Olympic programme. However, the presentation of judo as a coherent system of physical and moral education was made by its creator, Professor Jigoro Kanno, at the 1932 Olympic Games in Los Angeles during an occasional lecture [22].

Today, however, judo is primarily a well-known Olympic sport and practised by millions of people around the world. The International Judo Federation has 200 national federations spread across five continental associations (Africa, Asia, Europe, Oceania, and Pan-America). For decades, judo was the only Asian contribution to world sport. The cultural and preventive therapeutic phenomenon of not only judo but also other Asian combat sports was developed virtually only in Japan.

## 4. Therapeutic Perspective of the Numerous Possibilities of Hand-to-Hand Combat

### 4.1. Judo as a Modern Variation of Greek Wrestling and as an Inspiring Therapeutic Model

It is the founder of judo, Jigoro Kano, who can be considered a precursor of modern therapy and prevention using dynamic elements of martial arts. Safe fall (Japanese: ukemi waza) is a technique which teaches one how to protect the body against the destructive effects of a collision with the ground (prophylactic aspect), but in the case of people who have experienced the negative effects of a fall before, basic judo education and mastering safe fall techniques also has therapeutic functions. In a similar sense to judo education as a self-defence art (goshin jitsu no kata), those who have experienced the negative effects of physical violence can free themselves from this trauma. A set of many formal exercises (kata), due to the need to repeat the techniques to the right and left, symmetrically explores the muscles; the condition of prevention is met, but in many cases, it also provides possible therapy. Moreover, many Japanese universities have “Judo Therapy” faculties (the listed examples do not exhaust the preventive and therapeutic possibilities of judo).

Since 1989, as an element of cultural heritage, three traditional Japanese martial arts have been introduced into the curriculum of all types of schools under the common name of “budo”. Judo, kendo and sumo have become a compulsory part of children and young people’s education. “The dialogue between two minds and bodies in a face-to-face clash” in three different forms, including the use of symbolic incisive weapons from the earliest years of schooling, is a historical, cultural fact, promoting public health based on the heritage and wisdom of martial artists adapted to a civilised, modern society [23,24].

### 4.2. “The Dialogue between Two Minds and Bodies in a Face-to-Face Clash”—Still Untapped Therapeutic Potential of Martial Arts

The essential element that differentiates a group of combat sports within the Olympic family is their links to the philosophy and pragmatics of hand-to-hand combat. A characteristic feature of all of these disciplines is their complementary impact on the body of a young adept starting out in combat sports (i.e., on the somatic, motor, intellectual, moral and emotional spheres, etc.). It should be emphasised here that this possibility is rooted, on one hand, in the ancient slogan of the harmonious formation of spirit and body and, on the other hand, in the necessary competence of teachers. If these two criteria are fulfilled, among advanced athletes, mental development often becomes more important than the achievement of sporting success alone [25].

An expression of the expansion of combat sports and martial arts are the principles, general and specific methods, measures, tests, etc., which Kalina (2000) included in the theory of combat sports. Underpinning this unique theory are research findings that also demonstrate the potential for the philosophical–humanistic aspects of combat sports and martial arts to have a positive impact on contemporary society, particularly on individuals practising these forms of psychophysical activity. His classification of combat sports, taking into account the permissible ways of interacting with an opponent’s body, divides disciplines into three groups on the basis of the various forms of application of technical actions: weapons (fencing and kendo), striking (boxing, karate, taekwondo, etc.) and the use of throws and various techniques designed to ensure restraint of an opponent’s movements (judo, wrestling, etc.). He also points out the impact of martial arts and sports on the three personality spheres of people in training: mental, utilitarian and pragmatic [26]. This theory is an important element of the globally recognised new sub-discipline of science: the science of martial arts [27,28].

### 4.3. Therapeutic References to Hand-to-Hand Combat in the Modern Period Self-Defence

The first association relating to hand-to-hand combat is its very strong links with self-defence. The popularisation of this view in the West was crucially aided by numerous films in the hand-to-hand combat series featuring fighters played by actors of Asian origin. Since the end of World War II, martial arts films have become one of the pillars of Hong Kong popular culture and have become successful exports to Western countries. Behind the cinema screens were local practitioners who saw Chinese martial arts as essential skills and knowledge for self-defence, healthy living and character building. One cannot fail to point to the work of Bruce Lee in the USA and the establishment of numerous self-defence (karate) schools in the 1970s and 1980s around the world [29].

Self-defence is clearly perceived as an element of military and defence activities. Typically, an eclectic approach is used when designing self-defence systems. This indicates the use of various elements of martial arts in conjunction with other exercises involving general physical fitness development. This means that it is necessary to implement self-defence exercises with a comprehensive impact, taking into account mental development. In fact, the way it is applied is much more important than the technique itself [30].

Self-defence training as an extension of training systems related to sports and martial arts shapes, in general terms, life attitudes and self-esteem. Awareness of one’s own skills and the ability to act in the event of a threat to one’s own life or those of loved ones significantly increases self-esteem in the trainees.

This effect takes on additional significance in relation to women, who are largely portrayed as weak human beings compared to men in our society. The development of civilisation has changed this view, and the role of women in modern society includes all of the issues previously assigned only to men. Self-defence and martial arts exercises provide similar opportunities to pursue a life path for women and men. Self-defence training, through its impact, makes women the only decision-makers of their lives and their families. They can defend themselves against all adversities and are self-confident [31].

In fact, self-defence training can be carried out not only by able-bodied people, but also by people with disabilities, thanks to implemented scientific publications [32].

Widely understood defence education must certainly include physical exercises that includes elements of self-defence exercises and exercises with elements of combat sports or martial arts [33,34].

A unique situation regarding self-defence skills occurred during the 1984 European Wrestling Championships held in Jonkoping, Sweden. During the wrestling matches, the coach of the Polish national team, Stanisław Krzesiński, overpowered a terrorist.

The terrorist found himself at the sports facility, where he tried to disrupt the tournament by threatening to use a weapon. He also threatened to detonate a bomb that he supposedly had hidden in his suitcase. The coach of the Polish national team used his sports skills by making a quick attack on the opponent and restricting his movements, leading to the arrest of the terrorist [6].

### 4.4. From Reducing Violent Behaviour to Mental Health

Anger, verbal aggression and physical violence are obvious symptoms that indicate aggressive behaviour. The cognitive neo-associative model of aggression [35] presents the relationship between behaviours indicating aggression and specific external factors. It is these factors that give rise to other related ideas. The strength of the reaction depends on individual differences [36]. It can be pointed out that aggressive behaviour is determined by one’s own thoughts or decisions, which change their cognitive shape and shape their personality [37,38].

Combat sports can also be a cause of reducing aggressive behaviour if the intellectual, ethical and motor potential of the people involved in a direct confrontation is competently harnessed. Training and tournament fights, during which a kind of chivalric ritual applies, and ethical standards must be observed, are perhaps the most important means of harnessing this potential. The complexity of this type of interaction has led some researchers to analyse it in more detail. Underlying this approach is the real assumption that aggressiveness is not a fixed characteristic and that there is no empirical basis for separating this phenomenon into separate sets of women and men of similar age, occupational qualifications and socio-occupational activity.

Repeated circumstances of acute interpersonal conflict have a significant impact on the escalation or suppression (reduction) of aggression. Empirical justification for preventive and therapeutic agonology (in which self-defence exercises, fun forms of martial arts, judo fighting, etc. are key elements) was developed by Kalina. He defined two key phenomena related to human functioning in situations where interpersonal aggression occurs at the micro level (active counteraction: verbal, behavioural or mixed), and the potential lack of counteraction. These phenomena can be the basis for somewhat flexible recommendations for necessary and effective treatment and prevention [39,40]. 

In 2017, after a review of the literature examining the relationship between the level of aggression and success in sport [41], a higher level of aggression (actually aggressiveness) was found among players with less training experience. The authors emphasized that the results of research on sports aggression are often ambiguous, but this ambiguity may result from the inclusion of practitioners of various sports disciplines [42]. In conclusion, it was found that players with longer training experience are characterized by a lower level of aggressiveness [43,44,45,46,47].

In relation to martial arts coaches, covert aggression was found to be a better predictor of sporting performance than overt (verbal) aggression. The researchers indicated an increased awareness of the level of manifestation of various aggression factors in successful coaches; thus, this awareness prevented aggressive and unethical behaviour that would a negative impact on the wellbeing of (mainly) young athletes [48].

While the initial phase of study on aggressive behaviours focused primarily on those who participated in sports or martial arts, researchers are increasingly turning to target populations that have not been previously exposed to this type of physical exercise. The previously noticed impact of combat sports in reducing the level of aggression is now being addressed to young people in difficult social situations and exposed to aggressive behaviours in their environment. It was observed that proper dosing of this type of physical activity concentrates energy on positive patterns of activity. It was also emphasized that martial arts exercises create positive psychological changes. Of particular importance is the ability to self-control in school environments, where socialization behaviours are often disturbed and, as a consequence, exclusion occurs [49,50,51].

The basic element characterizing people participating in martial arts and sports exercises, according to the conducted research, is combining training programs with respect for fellow practitioners and the opponent during the fight. Thus, the requirement to control one’s own behaviour and the development of interpersonal respect is the main factor causing the reduction of aggression (aggressiveness), which also leads to an increase in self-awareness. Such responses are particularly important in the case of at-risk adolescents [52].

Martial arts training was found to have benefits related to improved cognition and increased self-confidence and self-esteem. A martial arts training program can be an effective sports-based intervention in improving mental health, positively affecting the resistance of students exposed to various factors of external and internal origin (e.g., those related to personality, perceived emotions, meeting needs, etc.) [53,54].

The range of influence through martial arts covers a growing area of intervention against bullying, harassment and other forms of violence in schools. The effects of a martial arts-based psychosocial intervention on high school students’ assessments of resilience and self-efficacy showed a consistent pattern of well-being outcomes [55].

### 4.5. Support for Medicine

The effects of the development of civilisation and technological progress, which is, unfortunately, also associated with the escalation of civilisation-related diseases, led to a sensible perception of the value of forms of physical activity in the areas of sports and martial arts not only with the stimulation of physical fitness in the general sense with self-defence skills, but also with the possibility of application in preventive health care and rehabilitation.

Nowadays, martial arts are practised both as a form of physical exercise without delving into their philosophical and moral aspects and as a sport with all of their negative consequences (injuries, years spent in disability and even death). However, widespread health-improving benefits were demonstrated in those actively involved in sports or martial arts, which can be expected. Advances in health behaviour research indicated that elements of these activities applied over a long period of time can significantly improve the wellbeing of their participants. It was found, for example, that judoists show a higher overall level of health-promoting behaviour than the average Polish person [56].

One of the first fields with a wide health impact that spread in Western civilisation was the Chinese martial art of tai chi. The effects of practicing this martial art and healing arts at the same time are documented in many scientific publications. Practiced as a specific form of therapy that combines functional and mental shaping, it has been known for over 300 years. The dissemination of this type of exercise in the entire part of the Asian continent and the different directions of development in its individual parts have led to the emergence of separate styles of Tai Chi, the impact of which combine movement, breath and mind. The balanced impact on the bodies of its practitioners as well as the psychological benefits resulting from the implementation of Tai Chi exercises were shown by numerous studies. It is the wide impact of Tai Chi exercises that makes them unique. In addition to metabolic changes, they positively affect the reduction of depression and perpetuate the improvement of depression indicators; thus, this confirms its mental health benefits [1,57,58,59,60]. The presented positive health effects of Tai Chi exercises were also noticed in relation to the elderly, reducing the frequency of falls and improving functioning of the immune system.

The use of Tai Chi exercises as a supplement to treatment is increasingly confirmed. It is one such approach to exercise that helps patients learn increasingly complex sequences of movements and improve the ability to combine individual movements with movements of the whole body [61,62].

Tai Chi exercises in the elderly may also improve brain function [63]. The conducted body of research showed an improvement in the ability to work in individual areas of the brain by improving the speed of processing the provided data. All of these activities lead to the possibility of more detailed work in the brain and beneficial changes in functional areas [64,65,66].

### 4.6. Boxing Exercise in Parkinson’s Disease

It is now widely accepted that combining interactions through physical activity with pharmacological intervention is a necessity to maximise their health-promoting effects in people with Parkinson’s disease. Physiotherapy and exercise can improve physical functioning. However, continuous exercise is needed to maintain results and promote a physically active lifestyle. Compliance with such long-term programmes remains a critical challenge. One possible solution to promote sustained exercise adherence is to participate in various non-traditional types of community-based group exercise programs [67,68,69,70].

Mobility in people with Parkinson’s disease (PD) is limited by poor coordination between body parts and between voluntary movements and associated postural adjustments, and by difficulties in switching motor programmes appropriate to changes in task limitations. A non-contact boxing programme called “Rock Steady Boxing” (RSB) is a popular exercise option for people with Parkinson’s disease (PD).

The combination of elements related to pharmacological actions, physical activity and external support through pro-health activities is a necessary condition to maintain the effects of therapy and shape the ability to lead a healthy lifestyle. The positive effects of therapy seem to be conditioned by the need to enable PD patients access to group exercises that are appropriately selected and with the longest possible duration. Various social groups respond to such demand by adapting forms of exercise characteristic of sports disciplines [67,68,69,70].

The specially developed exercise programme for people with PD includes complex, multi-segmented whole-body movements with the inclusion of tasks requiring rapid selection and sequencing of motor programmes, such as practising stance transitions (e.g., moving from stance to floor, rolling and standing up from floor to stance). Incorporating boxing actions into a memorised sequence is a way to practice rapid selection and sequencing of complex motor programmes for mobility. In order to address the problems associated with rapid programme selection, stride and agility exercises also provide practice in changing motor strategies when stopping, starting, changing direction, changing the walking limb and changing step size and placement [71].

Inability to perform a cognitive task and a balance or walking task simultaneously was found to be a predictor of falls in older people. It is even more difficult for a person with PD than age-matched older people to multitask, perhaps because the basal ganglia are responsible for automatic control of balance and gait and for switching attention between tasks. Posture increases most in people with PD who have a history of falls when a cognitive task is added to a silent posture task. These findings suggest that the ability to perform secondary cognitive or motor tasks while walking or maintaining balance is a key component of mobility that is particularly challenging in people with PD [72].

A specially developed exercise program includes complex, multi-segmented movements of the whole body, taking into account tasks that require appropriate selection and quick motor action (e.g., combining various simple actions into a complex whole). An important element of the program is the inclusion of selected boxing activities during the performance of simple motor activities, allowing for the development of coordination and the ability to quickly select a movement adapted to the required motor activity. All of these elements make it possible to maintain memorized simple and complex motor actions and force reactions to changes in the direction of movement [71].

Elderly people who additionally suffer from PD are at additional risk of injury as a result of falls. It has been shown that in the elderly, the main factor affecting the occurrence of falls is the impaired ability to perform a few simple motor activities, which results from difficulties in maintaining balance. Undoubtedly, the changes taking place in the brain as a result of the disease cause difficulties in sending the stimuli responsible for the control of balance and gait quickly enough. It was found that the transmission of this type of stimuli is significantly lower in people who have experienced falls [72].

Martial arts intervention in disease is not mainly limited to adult inflammatory, musculoskeletal or motor diseases, in which mechanical intervention results in positive changes. The effects of a defined martial arts intervention in children with cancer were also assessed in relation to their pain perception and coping. It was found that martial arts intervention may provide a useful method of reducing pain in childhood cancer, with greater effects achieved with higher baseline pain scores and patient age. A martial arts intervention can improve patient adherence to medical and surgical treatment, thereby reducing morbidity [73].

Research conducted among children with cancer showed new possibilities of using exercises with elements of martial arts. It was shown that this type of exercise can significantly reduce the level of pain experienced by these people. In addition, greater effects were found for higher initial pain values. Martial arts exercises can also improve medical compliance [73].

## 5. Conclusions

The exercise of hand-to-hand combat in its evolving forms has always been in the area of human interest. The essential element that distinguishes them from other forms of physical activity through this long period of time is the close link between their improvement of physical fitness and the formation of moral and ethical attitudes. This development was disrupted and forgotten for nearly 1500 years.

The revival of the modern Olympic Games through the introduction of Olympic ideals gave the Western world a new identity. In its reference, this ideal makes it possible to identify with this idea not only as a European idea, but also as a global idea, uniting the people and citizens of the world. In the first period of the modern era, sports and martial arts, through their strong links with elements of self-defence, accounted for the popularity of this type of physical activity. The second period saw the evolution of combat sports towards health-promoting behaviour. The effects of the development of civilisation, civilisational diseases and technological progress have led to a recognition of the value of forms of physical activity in the field of combat sports and martial arts not only in the development of physical fitness and the acquisition of self-defence skills, but also in the prevention of health problems and in rehabilitation. Advances in health behaviour research have indicated that elements of these activities applied over a long period of time can significantly improve the wellbeing of their participants. 

Nowadays, physical activity in the area of exercise with elements of hand-to-hand combat, combat sports or martial arts is beginning to form part of the prevention and treatment of 21st-century diseases. For Parkinson’s disease patients, drug treatment is the primary resource for continuing to function in society, but it will not be completely effective without the support of physical activity (e.g., “Rock Steady Boxing”). Of similar importance is the desire to prevent the occurrence of dangerous falls in the elderly or those with diseases related to the progress of civilisation, and special prevention programmes are increasingly being developed that include exercises with elements of hand-to-hand combat, combat sports or martial arts. Appropriate elements of this impact avoid the effects of lengthy treatment and even prevent injuries that would eliminate elderly people from public life [74,75,76,77,78,79,80,81].

Social health interventions currently focus on preventing dangerous situations much earlier. Teaching the young population to master the principles of safe falling significantly increases the likelihood of applying appropriate responses to falls in adulthood and old age. Actions that should be taken now for prevention can be realised through community programmes such as “Active today for a healthy future”.

## Data Availability

Not applicable.

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
