# Peer review of "From Ancient Patterns of Hand-to-Hand Combat to a Unique Therapy of the Future"

_ijerph, 2023, doi:10.3390/ijerph20043553_

Round 1

Reviewer 1 Report

Dear authors,

Thank you for writing this interesting and informative article. It is original through its use of Ancient Greek philosophical ideals as applied to contemporary martial arts, along with its use of the example of Rock Steady Boxing for people living with Parkinson's Disease. 

The main issue with the essay is a lack of a golden thread between the different sections. Judo appears in the middle, but then the shift in emphasis is on boxing, which could have its own section. The influence of Greek ideals of the body need to be examined in relation to the modern therapeutic modality, e.g. by adding information on the history of this therapy, its founder and their philosophy.

Below are some suggestions for the different sections of the essay, which, taken together, can make it easier to read and follow:

TITLE

This is quite long and confusing. "A therapy of the future" could read "a future therapy." Please consider revising this title.

ABSTRACT

- Was the Ancient Greek ideal just for men or women, too? What about older adults? Comments about this in the abstract and introduction would help indicate the relevance of these athletic ideals of beauty for an older, clinical population.

-"Will" has more of a Christian implication than an Ancient Greek one. This aspect of Christianity in modern Western civilisation also needs to be considered in the Introduction.

INTRODUCTION

- It is important to critique the idea that all martial arts have an Eastern origin.

- The male dominance of the Greek ideals could be expanded, e.g. from lines 102-104. Will modern women always relate to the likes of Hercules?

ROMANS AND MODERN OLYMPICS

- Christianity might be a unifying factor, as Europe changes from pagan societies to a monolithic one. 

- Disability sport, the Paralympics and the Special Olympics could be mentioned after the modern Olympic Games. 

- Line 234: Kano, not Kanno

- Kalina's taxonomy sounds interesting, although some martial arts actually include both weapons and striking and / or grappling. For instance, Aikido and Wing Chun also include weapons training.

- Line 271: Check for extra spacing

- Line 307: "the terrorist" should be "a terrorist"

- Line 309: "use weapon" should be "use a weapon"

- Line 383: Double full stop unnecessary

BOXING EXERCISE

- What does a traditional exercise programme normally consist of? Also check for spelling of program / programme to be consistent with your choice of English (US or UK).

- Check ordering of paragraphs in this section. Lines 440-441 could go with line 430.

- Rolling and moving to the floor are nor normal boxing techniques. Is there an influence from Judo in this regard?

- Lines 478-479 need a reference.

CONCLUSION

- Was such a holistic philosophy forgotten in the Western context? Eastern philosophy maintained the idea of cultivating body and mind to forge the character.

- Is the "Active today for a healthy future" slogan relating to Rock Steady Boxing? Could this boxing therapy become a preventative healthcare activity as well as a therapy for clinical patients with Parkinson's Disease?

Thanks in advance for considering this changes.

Author Response

Reply to the review

Dear reviewer,

Thank you very much for your time and thorough reading of the issues proposed by me, which brought very substantive comments to my study.

At the outset, I must point out that from the formal point of view, all the indicated comments have been corrected, which will certainly improve the value of this work, so I appreciate the reviewer's involvement in the analysis.

TITLE - shortened as suggested by the reviewer.

ABSTRACT - has been completed

INTRODUCTION - has been completed

ROMANS AND MODERN OLYMPIC GAMES - has been supplemented

BOXING EXERCISES

- What does a traditional exercise program usually consist of?

In this study, I wanted to draw attention to the fact of using forms of combat sports exercises, martial arts, associated with people with very high physical fitness, as elements supporting the processes of therapy or rehabilitation. My intention was to show other disciplines the possibility of positively influencing society through exercises with elements of combat sports, appropriately adapted to the needs of the elderly.

CONCLUSION

- Has such a holistic philosophy been forgotten in the Western context? Eastern philosophy held the idea of cultivating body and mind to forge character.

This issue has been noticed by me and I want to develop it in my further work. The main element discussed there is an indication of the reasons for the fall of the principles shaped by the Greeks and referring them to the philosophy of the East. An indication of the historical conditions of the brutalization of forms of struggle and the departure from the principles of noble combat in the Roman period with modern changes proposed to the present Western societies. In conclusion, the question must be asked: can we make the same mistakes?

- does the slogan "Active today for a healthy future" refer to Rock Steady Boxing? Can this boxing therapy become a health prophylaxis as well as therapy for clinical patients with Parkinson's disease?

Based on the research (included in the work) indicating the effectiveness of this therapy, with particular emphasis on the mutual complementation of pharmacological therapy and therapy related to physical activity (e.g. through Rock Steady Boxing), I suggest that representatives of other martial arts should cooperate with physicians in order to develop appropriate forms of physical activity.

The slogan "Active today for a healthy future" is one of the proposals of EU social programs on the basis of which the tasks of improving the physical activity of modern societies can be implemented. Of course, to a lesser extent, it may refer to the Rock Steady Boxing prophylaxis already implemented, and more to the prophylaxis of preventing falls by teaching safe falling techniques to young people, which can reduce this type of injury in adulthood.

Once again, thank you for your thorough review, I hope that the corrections and clarifications will be sufficient.

The changes in the manuscript are highlighted in red colour

Reviewer 2 Report

An interesting popular science article that can serve as an introduction or research inspiration for other authors. I miss a bit of an indication of the explicit purpose of the article, the setting of some hypothesis.

I did not notice a research question in the paper. It's more the form of a review or popular science article.

The article is not original, but may be relevant to the field, as not many people notice that practicing martial arts can be part of preventive health care.

The article can provide source material. It expands the available literature.

The article is a popular science review.

The references are appropriate.

Author Response

Reply to the review

Dear reviewer,

Thank you very much for your time and thorough reading of the issues proposed by me, which brought very substantive comments to my study.

At the outset, I must point out that from the formal point of view, all the indicated comments have been corrected, which will certainly improve the value of this work, so I appreciate the reviewer's involvement in the analysis.

TITLE - shortened as suggested by the reviewer.

ABSTRACT - has been completed

Once again, thank you for your thorough review, I hope that the corrections and clarifications will be sufficient.

Round 2

Reviewer 1 Report

Dear Author,

Many thanks for returning this revised manuscript in such a prompt and effective manner. Your reflective response in the letter is also appreciated. I have read through the second version of the paper, which now makes perfect sense. You have addressed my main concerns through detailed and thoughtful paragraphs, which help link different ideas raised in the article. Now the manuscript is ready for publication, so thank you once again.

Kind regards.